# Bridging disparity in knowledge and utilization of contraceptive methods among married couples in the pastoralist community of Fentale District, Eastern Ethiopia

**Sena Adugna Beyene**[1,2]*, **Sileshi Garoma**[3], **Tefera Belachew**[4]

1 Department of Population and Family Health, Institute of Health, Jimma University, Jimma, Ethiopia, 2 Department of Statistics, College of Natural Sciences, Jimma University, Jimma, Ethiopia, 3 Departments of Public Health, Adama Hospital Medical College, Adama, Ethiopia, 4 Department of Nutrition & Dietetics, Institute of Health, Jimma University, Jimma, Ethiopia

* senaada491@gmail.com

## Abstract

### Background

Despite progress in national reproductive health, pastoralist regions, including the Fentale District in Eastern Ethiopia, face challenges with limited contraceptive coverage due to gaps in understanding and adoption among couples.

### Methods

This cross-sectional study of 1496 couples in Fentale District, Eastern Ethiopia, from October 1 to December 25, 2021, utilized multistage sampling. Data entered into EPI Data underwent analysis with SPSS (v23.0) and STATA (v14.0), employing descriptive statistics, bivariate analysis, and binary logistic regression to identify predictors of contraceptive knowledge.

### Results

Of the 1496 couples, 93.8% (1404) participated in the survey. Significant educational disparities were observed, with 53.8% having no formal education, particularly among women (65.2%). Despite this, 80.2% of couples were aware of at least one family planning (FP) method, and 78.6% knew modern methods. The median knowledge of contraception methods was 3 for both genders. Men showed higher awareness of male (43.2%) and female condoms (17.4%) compared to women (31.8% and 6.0%). Overall, 27.4% of couples used contraception, with a significant gender difference: 41.2% of women and 13.5% of men. Modern contraceptives were used by 18.2% of couples, predominantly by women (34.8%). Binary logistic regression analysis indicated positive associations between contraceptive knowledge and education, bank account ownership, occupation, proximity to healthcare, and media exposure, while a nomadic-pastoralist lifestyle and specific treatment preferences showed negative associations.

**Data Availability Statement:** The paper's main text presents the primary results and analyses, including all crucial data points, statistical analyses, and relevant figures or tables necessary for

understanding and validating our findings. For further information or specific data inquiries, readers are encouraged to contact the Institutional Review Board of Jimma University at ero@ju.edu. et, as provided in the paper.

**Funding:** The author(s) received no specific funding for this work.

**Competing interests:** The authors have declared that no competing interests exist

## Conclusion

The limited knowledge and utilization of family planning in the Fentale District stem from gaps in comprehension and disparities among couples. Factors influencing this situation include socio-demographic considerations, such as variations based on education, occupation, media exposure, bank account ownership, treatment preferences, and distance from healthcare facilities. This ensures that the interventions are having the desired effect and allows for adjustments as needed to promote family planning uptake.

## 1. Introduction

Reproductive health and family planning are critical components of global health initiatives, aiming to achieve the Sustainable Development Goals (SDGs), particularly SDG 3 on good health and well-being [1]. Globally, there is a concerted effort to improve access to family planning services, yet underserved communities, such as nomadic pastoralists, continue to face significant challenges [2].

In Sub-Saharan Africa, pastoralist communities encounter unique obstacles to accessing essential services, including family planning [3]. Ethiopia's Health Sector Transformation Plan II (HSTP II) prioritizes comprehensive family planning services as a key strategy to improve reproductive health outcomes [4].

The Fentale District in Eastern Ethiopia is home to a nomadic pastoralist community that faces considerable challenges in accessing essential services due to their mobile lifestyle [5]. Previous studies in South Omo, Ethiopia, and Chad have underscored the difficulties in providing services to such communities, highlighting the need for tailored approaches [6, 7].

Limited adoption of contraceptive methods among couples refers to the lower rates at which married couples in pastoralist regions, such as the Fentale District in Eastern Ethiopia, choose to use or accept family planning methods [8]. This low adoption can result from various factors, including cultural beliefs, lack of access to information or services, misconceptions about contraceptives, and preferences for larger families [3]. When couples do not adopt or utilize contraceptives effectively, it leads to gaps in contraceptive coverage within the community [9].

These gaps contribute to higher fertility rates, unintended pregnancies, and challenges in achieving reproductive health goals at both individual and community levels [8]. Addressing these adoption barriers is crucial for improving contraceptive coverage and promoting reproductive health in pastoralist communities [10]. This study focuses on the reproductive health dynamics in the Fentale District, emphasizing the crucial role of both men and women in family planning decision-making [11]. The objectives include exploring modern contraceptive use, contraceptive knowledge [12], and the socio-demographic factors influencing family planning practices among married couples in the district [11].

Addressing the challenges faced by pastoral communities in accessing essential services, this study advocates for tailored health education programs for both men and women, recognizing the pivotal role of both genders in reproductive health decision-making [13]. It also emphasizes engaging community leaders, religious figures, and "Abba Gada" (Indigenous Oromo) to promote family planning and ensure culturally sensitive [14–16]. Additionally, the study recommends implementing mobile clinics and adaptive social services to reach nomadic populations effectively [7, 17].

This study contributes to the global discourse on family planning and reproductive health, offering valuable insights for family planning program managers and service providers [18]. The findings aim to inform tailored strategies that can enhance family planning outcomes and reproductive health in pastoralist regions, ultimately supporting Ethiopia's growth transformation plan and the attainment of sustainable development goals [4]. This study aims to identify disparities in knowledge and utilization of contraceptive methods among married couples in this community and propose potential interventions to address these gaps.

## 2. Methods and materials

### 2.1. Study area

The research is conducted in Fentale Woreda, located within the East Showa zone of the Oromia regional state in the southern part of the northern rift valley of Ethiopia. The area is characterized by a nomadic lifestyle, agro-pastoralism, and seasonal migration patterns typical of pastoralist communities. Livestock production is central to the local economy.

Fentale district, part of the East Showa Zone in Fantalle Woreda, encompasses 20 kebeles, including 18 rural and 2 urban administration kebeles, with 15 specifically designated as pastoralist kebeles. The study focuses on pastoralist villages such as Kobo, Benti, Gola, Dhaga Edu, Tututi, Ilalla, and Gelcha. Healthcare infrastructure includes four health centers, each village with a health post, and an additional health center in Metehara City. The hospital primarily serves non-pastoralists and employees of the Metehara sugar factory, posing challenges for pastoralists due to limited transportation options for accessing family planning services provided exclusively at the Metehara Hospital and Health Center.

Accessing healthcare services requires hours or a full day of walking, causing significant delays as pastoralists await transportation, sometimes up to a week. Health centers are predominantly staffed by nurses, while Health Extension Workers (HEWs) manage health posts. Traditional birth attendants, locally known as "Deesisttu Aadaa," play a crucial role in childbirth assistance and hold considerable respect within the community. The social, economic, and cultural aspects of the study area profoundly influence its unique context, providing the backdrop for this research. For further details on these aspects, please refer to the comprehensive description in the cited sources, offering a holistic understanding of the study setting [19, 20].

### 2.2. Study design, time frame, and sampling approach

Our study employed a multi-stage sampling strategy, using districts (woredas) as primary sampling units (PSUs) and sub-districts (kebeles) as secondary sampling units (SSUs). Fentale district, located in the East Showa Zone, encompasses 20 kebeles, of which 18 are rural and 2 are urban administration kebeles. Initially, 15 pastoralist kebeles were purposefully selected based on various criteria, including accessibility, social structure, economic strength, and pastoralist nature. Subsequently, seven kebeles were randomly chosen from this initial selection. The source population consisted of married couples, systematically sampled from the total women of reproductive age in each kebele. Systematic sampling ensured representative selection, with villages chosen based on proximity to health facilities to minimize variations. In the seven pastoralist sub-districts, a total of 1,045 women of reproductive age and their husbands (2,090 married couples) resided in 5,223 households. The allocation of the total sample size was based on the probability proportion to the size of the selected pastoralist sub-districts, determined by the number of households in each kebele. Within each sub-district, married couples were selected for interviews at equal intervals using a systematic sampling technique as shown previously in [8] (refer to Fig 1). Fieldwork for this couple-based cross-sectional study was conducted between October 1 and December 25, 2021.

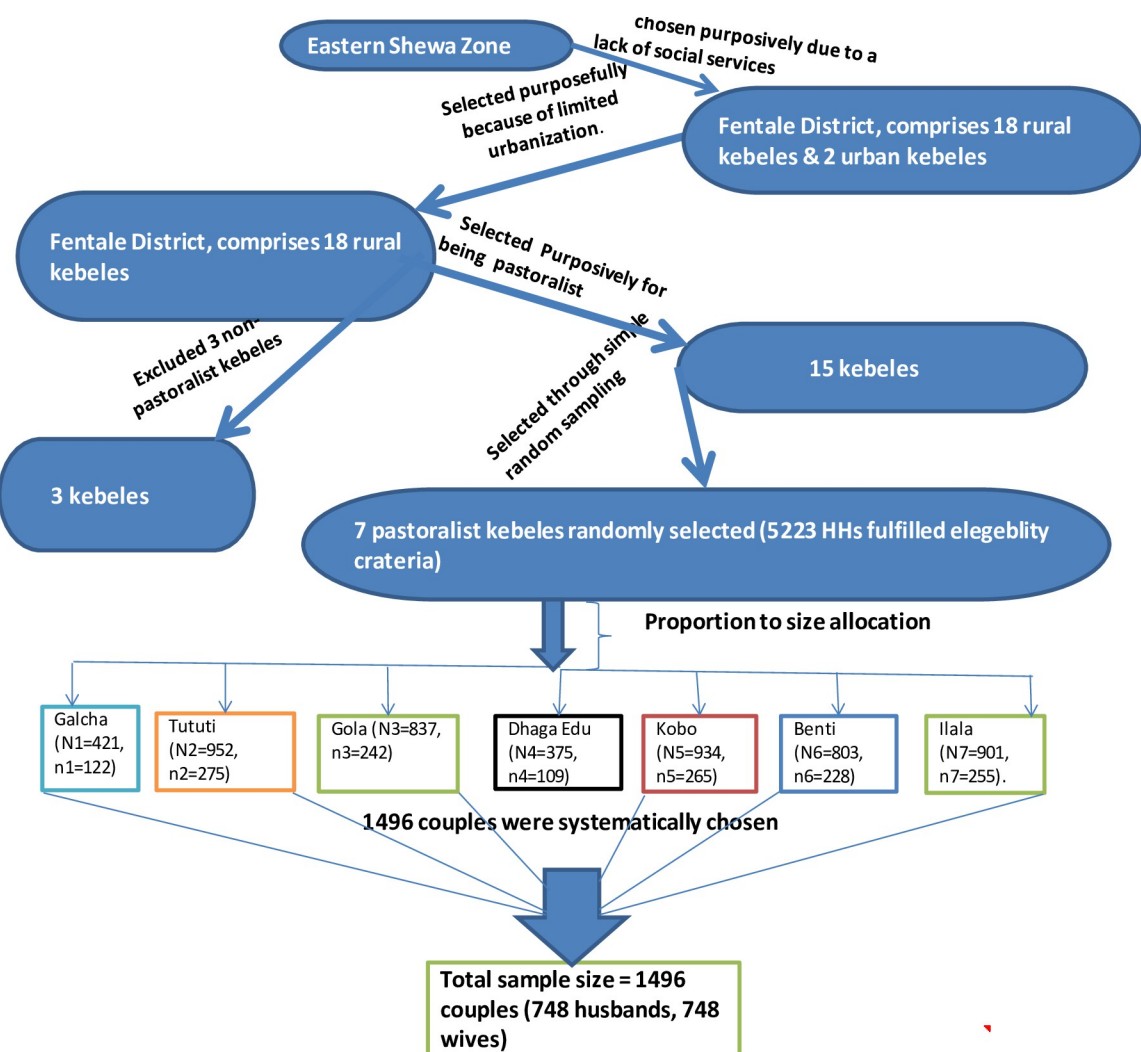

**Fig 1. Illustrates the schematic presentation of the sampling procedure conducted among married couples in the pastoralist community of Fentale District, Eastern Ethiopia, from October 1 to December 25, 2021.**

## 2.3. Study population

This study focused on the pastoralist community residing in Fentale Woreda, located within the East Showa zone of the Oromia regional state, Ethiopia. The target population was carefully defined with specific inclusion and exclusion criteria to emphasize the unique characteristics of the participants.

## 2.4. Source population

The source population for this study is the pastoralist community residing in Fentale Woreda, Oromia regional state, Ethiopia. This community is characterized by its nomadic or semi-nomadic lifestyle, primarily engaged in livestock herding and moving in search of grass and water for their animals.

The study population was specifically selected from the source population based on the following inclusion and exclusion criteria:

The study outlined its target population, establishing precise criteria for inclusion and exclusion to emphasize the unique characteristics of the participants. Inclusion criteria encompassed: married women aged 15 to 49 and their husbands; non-pregnant women and their husbands; Legally wedded couples residing in the village or an area with consistent mobility for at least a year in search of grass and water for livestock; couples cohabiting within the study area or in mobile regions; couples committed to remaining in the district or mobile areas for at least a year and a half from the data collection period; husbands consenting for their wives' participation; mentally competent couples; and husbands in monogamous marriages. For wives under 18, written informed consent was obtained from their husbands, respecting cultural norms. Exclusions included: married women and their husbands outside the 15 to 49 age range; couples not legally married or where the husband declined to include his wife; mentally incapacitated couples; husbands in polygamous marriages; pregnant women and their husbands; couples not residing in the village or mobile areas for the past year; couples not cohabiting in the same household in the study area or mobile regions; and couples not planning to stay in the district or mobile areas for at least a year and a half from the time of data collection. These exclusions were designed to maintain focus and minimize redundant information.

## 2.5. Sample size determination

The sample size determination was guided by specific parameters outlined in [8]. This involved calculation incorporating a standard normal deviation (Z) set at $Z\alpha/2 = 1.96$ for a 95% confidence interval (CI) and assumed 90% power.

The estimated key proportion of current family planning utilization in the Oromia regional state, based on the EDHS (2016) report [21], served as a basis for the calculation (28.1% or 0.281). Accounting for a desired precision of 0.05, a finite population correction, and an Intra-Cluster Correlation Coefficient (ICC) variation of 0.05 for clustering effects. Initially, the sample size was determined to be 310 couples, with adjustments introducing a 20% increment to compensate for expected non-responses. Additionally, a design effect of 2.2 was applied. Consequently, the final sample size was determined to be 374 couples, resulting in a total of 748 couples per group selected using systematic sampling. The cumulative sample size for the study thus comprised 1496 couples. This systematic approach ensures the robustness and statistical significance of the study findings.

## 2.6. Data collection

The structured and pre-tested questionnaire, originally developed in English, was meticulously translated into Oromiffa, the local language, and then back-translated into English to ensure the questions' consistency and accuracy. Tailored versions of the questionnaire were administered to male and female respondents, covering various aspects such as socio-demographic and economic factors, reproductive history, and knowledge and practices concerning family planning (FP). The questionnaire delved into topics such as past and current contraceptive use, reasons for non-use, types of contraceptives, awareness of usage, sources of family planning services, and potential side effects. The survey instruments were adapted from a validated questionnaire, deemed valid and reliable by experts [8, 11, 21–26].

Before the main data collection phase, the questionnaire was pilot-tested with 5% of couples from a different district to evaluate its validity and clarity. The study team consisted of 15 male and 15 female data collectors, overseen by six field coordinators, all of whom were recruited from the local community. Data collectors were matched with respondents of the same gender to address the sensitivity of the topics discussed. Interviews were conducted privately with

each couple, respecting confidentiality and choosing locations based on the participants' comfort.

All data collectors and supervisors completed a comprehensive six-day training program that covered study objectives, procedures, data collection methods, interviewing techniques, and addressed any concerns. The training included practical exercises, such as role-playing, to enhance skills. To ensure consistency and comparability, both men and women were given equivalent questionnaires, allowing for direct and accurate comparisons in the analysis.

Participants were given detailed information about the study, including its goals, procedures, potential risks, and benefits, to ensure informed consent. The consistent approach to questionnaire administration across genders, combined with the analytical method chosen, ensured a robust analysis that considered within-couple dynamics and utilized culturally appropriate data collection instruments.

## 2.7. Pastoralism in Ethiopia

The essence of pastoralism in Ethiopia embodies both a cultural and livelihood system deeply intertwined with the vast rangelands. This lifestyle is especially prominent in underdeveloped areas marked by scarce social services and infrastructure. The critical production system plays a vital role in arid and semi-arid dry land areas [27]. The differentiation between nomadic pastoralism and agro-pastoralism is based on the primary source of income. Nomadic pastoralism revolves around livestock and their products, featuring mobility for grazing and water, while agro-pastoralists mainly concentrate on cultivation with less focus on livestock production, living permanently in their respective areas [28]. Challenges faced by pastoralists, including high population growth, long-standing resource conflicts, and restricted access to grazing lands and water sources, are often associated with climate change and widespread animal diseases [27].

## 2.8. Measures

**2.8.1. Knowledge of family planning (FP).** Eleven questions about types of contraceptive methods were posed to evaluate participants' knowledge. Responses of "Yes" were coded as "1" and "No" as "0" for each question. Subsequently, a knowledge score was calculated for each participant, ranging from 0 to 11. The score's normality was then checked, and a composite knowledge variable was created using the mean as the cutoff score. Participants with scores at or above the mean were classified as "knowledgeable," while those below the mean were classified as "less knowledgeable." Additionally, internal consistency was meticulously measured to gauge reliability. Cronbach's Alpha ($\alpha$) values indicated excellent internal consistency reliability for the knowledge of contraceptive methods scale employed in the study. The alpha ($\alpha$) values for the 11 knowledge-related items were exceptionally strong, with an $\alpha$ of 1.000.

**2.8.2. Current Use of family planning.** In our study, "current use of family planning" refers to participants actively using a family planning method, either by themselves or through their partners, during the data collection period.

## 2.9. Data management and analysis approach

In the Data Management and Analysis Approach, rigorous procedures were followed to ensure the accuracy and reliability of the collected data. Upon completion of questionnaire responses, each was carefully reviewed for completeness and assigned unique codes. Data entry was performed using a validated template, which underwent thorough validation with 30 questionnaires overseen by the principal investigator. Subsequently, the data were transferred to statistical software packages, including SPSS version 23.0 and STATA version 14.0, for comprehensive analysis.

By combining the strengths of both SPSS and STATA, we conducted a thorough and detailed analysis, ensuring our findings were accurate and comprehensive. The use of SPSS and STATA in tandem provided a balanced approach to data analysis. SPSS is renowned for its user-friendly interface and robust descriptive statistics capabilities, making it ideal for initial data exploration and summary statistics. STATA, on the other hand, enables more complex statistical modeling and handles large datasets efficiently.

For more complex analyses, such as bivariate analysis and binary logistic regression to identify predictors of contraceptive knowledge, we used STATA. Its advanced statistical functions and ability to handle large datasets provided more precise and reliable results. This complementary use of both software programs enhanced the overall rigor and robustness of our research.

Frequency analyses were conducted to examine event occurrences, and specialized tests such as Pearson's Chi-square were utilized to determine significance. Associations between family planning knowledge, contraceptive use, and various factors were explored using Pearson's Chi-square and logistic regression. We addressed potential multicollinearity among all independent variables using VIF analysis, confirming data integrity with no significant issues. This assessment is crucial for our analysis of contraceptive knowledge, ensuring that correlations between variables remain manageable. Only the significant predictors with a probability value of less than 5% were included in the multivariate logistic regression to assess the relationship with the outcome by controlling for other variables in the model.

Multivariate binary logistic models were employed to predict factors influencing knowledge, reporting odds ratios with confidence intervals for nuanced interpretation. Univariate analysis provided insights into the distribution of individual variables, while descriptive statistics summarized variable characteristics. Bivariate analysis examined relationships between variables through cross-tabulations and correlation analyses, identifying associations and dependencies. Crude odds ratios were calculated to gauge the association between individual independent variables and dependent variables, with adjusted odds ratios derived from multivariate binary logistic models. The study also considered survey design effects to ensure analytical validity. Results were meticulously reviewed to confirm the effectiveness of the chosen methods in addressing research questions and objectives while adapting to changing data characteristics.

## 2.10. Ethical consideration

All experimental protocols for this study were approved by the Institutional Review Board of Jimma University and the Oromia Regional Health Bureau before the study commenced. Ethical clearance was granted by the Institute of Health Institutional Review Board at Jimma University (IHRPG-927/2020) and the Health Ethical Review Committee of the Oromia Regional Health Bureau (BEFO/HBTFH/999/2020). Additionally, formal consent was obtained from the Health Department of the Eastern Shewa Zone in the Oromia regional state and Fentale District within the Eastern Shewa Zone. Before participating, all respondents gave informed, voluntary, verbal consent after receiving detailed explanations about the study's goals and methods. The research strictly followed fundamental principles of human research ethics, including respect for individuals, beneficence, voluntary participation, confidentiality, and justice. Moreover, verbal informed consent was acquired from spouses acting on behalf of wives under 18, in accordance with cultural norms and ethical guidelines. No direct compensation was offered, and all procedures were conducted in compliance with the regulations set forth by the ethics committee.

## 3. Results

### 3.1. Socio-demographic and reproductive disparities

This study, conducted in the Fentale District, Eastern Ethiopia, aimed to explore family planning knowledge and contraceptive among pastoralist communities. A total of 1496 eligible married couples were identified, with 93.8% (1404) participating in the cross-sectional survey. The nomadic pastoralist lifestyle, characterized by frequent relocations for livestock needs, posed challenges in participant engagement. Women aged 15–49 had a median age of 26 (IQR = [21; 30]), while men had a median age of 30 (IQR = [26; 40]). Median ages at first marriage were 18 years [IQR = 16; 19] for men and 15 years [IQR = 14; 18] for women (see Table 1), highlighting age variations and marital trends among genders. An analysis of respondents revealed significant educational disparities: 53.8% had no formal education, with a higher percentage among women (65.2%) compared to men (42.5%). Conversely, 32.6% had completed primary education, with more men (38.0%) than women (27.2%). Moreover, 13.5% of couples had secondary education or higher, predominantly men (19.5%) compared to women (7.5%). Ethnically, 99.6% identified as Oromo, and 97.9% were Muslim, reflecting the demographic composition of the pastoral kebeles.

The primary occupation for 64.6% of couples was nomadic pastoralism, slightly higher among women (62.4%) than men (66.8%). Agro-pastoralism was adopted by 24.3% of couples, with similar representation between genders. Media exposure was low, particularly among women (93.3%), and illiterate individuals had less exposure.

Regarding travel patterns, 27.6% reported separate travels three times a year, with men slightly more frequent. Median migration frequency was 4 [IQR = 3; 4], higher among men (4 [IQR = 3; 5]) than women (3 [IQR = 2; 4]).

Disparities in ownership of mobile phones, radios, and bank accounts were evident, favoring men. Internet usage was minimal, more among men (8.1%) than women (3.6%). Migration within Fentale District was common (85.9%), especially among women (93.0%), with 14.1% migrating outside the district, more men (21.2%) than women (7.0%).

Healthcare preferences varied: 32.6% of men preferred health sectors for treatment, while 43.4% opted for traditional healers, more women (48.9%) than men (37.7%). Religious leaders were preferred by 31.7%, slightly more among women (35.0%).

Access to healthcare involved significant walking time, with 57.5% walking ≥1 hour, more women (71.2%) than men (43.7%). Family planning discussions were infrequent (93.2%), more among women (94.9%) than men (91.5%).

Approximately 42.5% received family planning information from health extension workers, underscoring their crucial role as information sources. Moreover, 44.4% demonstrated comprehensive family planning knowledge, likely influenced by health extension workers. Median household size was 5 (IQR = [4; 7]), with a desire for 3 additional children, smaller in literate households (refer to Table 1 for details).

### 3.2. Knowledge of family planning methods

Concerning family planning awareness, 80.2% of couples demonstrated knowledge of at least one family planning (FP) method, with 78.6% aware of modern FP methods. The median knowledge of contraception methods for both men and women was 3, with an average of 3.12. No statistically significant difference in knowledge was observed between genders (P = 0.060). However, there were gender disparities in awareness of specific FP methods; men showed higher knowledge of male condoms (43.2%) and female condoms (17.4%) compared to women (31.8% for male condoms, 6.0% for female condoms). Overall, couples' awareness of these

**Table 1. Socio-demographic characteristics and reproductive history difference among married couples, Fentale Districts, Eastern Ethiopia.**

| Distribution (%) | | | | |
|---|---|---|---|---|
| Characteristics at individual level | Women (N = 702) | Men (N = 702) | Total (N = 1404) | P-value |
| Median age at first Marriage | 15[IQR = 14;18] | 18[IQR = 16;19] | 17[IQR = 14;19] | NA** |
| Median age | 26[IQR = 21;30] | 30[IQR = 26;40] | 28[IQR = 23;34] | NA** |
| Age | | | | 0.000* |
| 15–19 years | 94(13.4) | 15(2.1) | 109(7.8) | |
| 20–24 years | 186(26.5) | 122(17.4) | 308(21.9) | |
| 25–29 years | 202(28.8) | 170(24.2) | 372(26.5) | |
| 30–34 years | 159(22.6) | 121(17.2) | 280(19.9) | |
| 35–39 years | 38(5.4) | 81(11.5) | 119(8.5) | |
| 40–44 years | 12(1.7) | 105(15.1) | 117(8.3) | |
| > = 45 years | 11(1.6) | 88(12.5) | 99(7.1) | |
| Educational status | | | | 0.000* |
| No formal education | 458(65.2) | 298(42.5) | 756(53.8) | |
| Primary | 191(27.2) | 267(38.0) | 458(32.6) | |
| Secondary & above | 53(7.5) | 137(19.5) | 190(13.5) | |
| Religion | | | | 0.712 |
| Muslim | 686(97.7) | 688(98.0) | 1374(97.9) | |
| Christian | 16(2.3) | 14(2.0) | 30(2.1) | |
| Ethnicity | | | | 1.000 |
| Oromo | 699(99.6) | 699(99.6) | 99.6 | |
| Others | 3(0.4) | 3(0.4) | 6(0.4) | |
| Occupational status | | | | 0.000* |
| Nomadic-pastoralist | 438(62.4) | 469(66.8) | 907(64.6) | |
| Business | 75(10.7) | 25(3.6) | 100(7.1) | |
| Others | 8(1.1) | 17(2.4) | 25(1.8) | |
| Student | 8(1.1) | 23(3.3) | 31(2.2) | |
| Agro-Pastoralist | 173(24.6) | 168(23.9) | 341(24.3) | |
| Possession of radio | | | | 0.021* |
| No | 672(95.7) | 652(92.9) | 1324(94.3) | |
| Yes | 30(4.3) | 50(7.1) | 80(5.7) | |
| Possession of mobile phone | | | | 0.000* |
| No | 615(87.6) | 534(76.1) | 1149(81.8) | |
| Yes | 87(12.4) | 168(23.9) | 255(18.2) | |
| Possession of Bank account | | | | 0.000* |
| No | 657(93.6) | 160(22.8) | 817(58.2) | |
| Yes | 45(6.4) | 542(77.2) | 587(41.8) | |
| Use the internet | | | | 0.000* |
| No | 677(96.4) | 645(91.9) | 1322(94.2) | |
| Yes | 25(3.6) | 57(8.1) | 82(5.8) | |
| Couple's Exposure to media | | | | 0.000* |
| Less Frequent | 655(93.3) | 568(80.9) | 1223(87.1) | |
| More Frequent | 47(6.7) | 134(19.1) | 181(12.9) | |
| Frequency of Migration | | | | 0.000* |
| Once | 18(2.6) | 3(0.4) | 21(1.5) | |
| Twice | 185(26.4) | 62(8.8) | 247(17.6) | |
| Three | 187(26.6) | 201(28.6) | 388(27.6) | |
| Four | 224(31.9) | 259(36.9) | 483(34.4) | |
| Five& more | 88(12.5) | 177(25.2) | 265(18.9) | |

(*Continued*)

**Table 1.** (Continued)

| Distribution (%) | | | | |
|---|---|---|---|---|
| **Characteristics at individual level** | **Women (N = 702)** | **Men (N = 702)** | **Total (N = 1404)** | **P-value** |
| Median migration frequency | 3[IQR = 2;4] | 4[IQR = 3;5] | 4[IQR = 3;4] | NA** |
| Migration destination | | | | 0.000* |
| Within Fentale District | 653(93.0) | 553(78.8) | 1206(85.9) | |
| Outside Fentale District | 49(7.0) | 149(21.2) | 198(14.1) | |
| Family structure who migrate mostly | | | | 0.208* |
| Head of the household | 324(46.2) | 318(45.3) | 642(45.7) | |
| All family members | 205(29.2) | 184(26.2) | 389(27.7) | |
| Young Men | 173(24.6) | 200(28.5) | 373(26.6) | |
| Treatment Seeking | | | | 0.000* |
| At Health Sectors | 102(14.5) | 229(32.6) | 331(23.6) | |
| Traditional healers | 343(48.9) | 265(37.7) | 608(43.3) | |
| Religious places | 246(35.0) | 199(28.3) | 445(31.7) | |
| Others | 11(1.6) | 9(1.3) | 20(1.4) | |
| Distance from health center | | | | 0.000* |
| < 1 hour | 202(28.8) | 395(56.3) | 597(42.5) | |
| ≥ 1 hour | 500(71.2) | 307(43.7) | 807(57.5) | |
| Couple discussion of FP | | | | 0.011* |
| Never discussed | 666(94.9) | 642(91.5) | 1308(93.2) | |
| Discussed | 36(5.1) | 60(8,5) | 96(6.8) | |
| Family size | | | | 0.905 |
| < = 4 people | 269(38.3) | 265(37.7) | 534(38.0) | |
| 5–8 people | 361(51.4) | 360(51.3) | 721(51.4) | |
| > = 9 people | 72(10.3) | 77(11.0) | 149(10.6) | |
| Median family size | 5[IQR = 4;7] | 5[IQR = 4;7] | 5[IQR = 4;7] | NA** |
| Desired number of children | | | | 0.000* |
| 0 | 170(24.2) | 64(9.1) | 234(16.7) | |
| 1–2 | 115(16.4) | 111(15.8) | 226(16.1) | |
| 3–5 | 250(35.6) | 301(42.9) | 551(39.2) | |
| >5 | 167(23.8) | 226(32.2) | 393(28.0) | |
| Median desired number of children | 3[IQR = 1;5] | 4[IQR = 3;6] | 4[IQR = 2;6] | NA** |
| Need for future child | | | | 0.000* |
| No | 170(24.2) | 64(9.1) | 234(16.7) | |
| Yes | 532(75.8) | 638(90.9) | 1170(83.3) | |

*Implies statistically significant results at 5% level of significance.

Note.NA**: Not Applicable

methods was 37.5% for male condoms and 11.7% for female condoms (p = 0.000). Knowledge of male condoms was greater for both males and females than for female condoms. Significant gender differences were noted for permanent methods ($\chi 2$ (1, N = 1404) = 4.952, p = 0.026).

Generally, there was no statistically significant difference in knowledge among spouses regarding short-term methods (pills, injections), long-term methods (IUD, implants), and natural methods (lactational amenorrhea, periodic abstinence, withdrawal). The most recognized contraceptive methods were pills (74.7%), injectables (72.7%), and implants (39.0%). The findings revealed higher knowledge about short-term methods (78.6%) than long-term methods

(41.5%), natural methods (40.3%), and permanent methods (4.3%). Couples generally exhibited low knowledge of permanent methods (4.3%) compared to other contraceptive methods. (Refer to Table 2 for details).

### 3.3. Contraceptive knowledge and utilization

Among married couples, 27.4% reported using some form of contraception, revealing a significant difference in usage between women (41.2%) and men (13.5%). When evaluating knowledge related to contraceptive use among couples who employ any form of contraception, 46.9% showed awareness, with a substantial difference between women (61.6%) and men (23.7%).

Regarding the use of modern contraception, 18.2% of married couples reported its use, with a notable difference observed between women (34.8%) and men (1.7%). Analyzing the knowledge level of couples utilizing modern contraception, 24.4% demonstrated awareness, highlighting a significant difference between women (49.7%) and men (1.8%).

In terms of overall contraceptive knowledge, 44.4% of married couples had knowledge, indicating a noteworthy difference between women (41.9%) and men (46.9%). (Refer to Figs 2 and 3 for details).

### 3.4. Couples' disparities in sources of information on contraceptive methods

Approximately 42.5% of couples obtained information about family planning, with health extension workers being a significant source. Health extension workers contributed to the family planning knowledge of 42% of husbands and 43% of wives. Mass media (TV/radio/

**Table 2. Knowledge difference of contraceptive methods among married couples Fentale Districts, Eastern Ethiopia.**

| Distribution (%) | | | | |
|---|---|---|---|---|
| **FP methods use** | **Men (N = 702)** | **Women (N = 702)** | **Total (N = 1404)** | **P-value** |
| Ever heard of any family planning method | 565(80.5) | 561(79.9) | 1126(80.2) | 0.789 |
| Ever heard of any modern family planning method | 552(78.6) | 552(78.6) | 552(78.6) | 1.000 |
| Aware of all modern FP methods | 2(0.3) | 0(0.0) | 2(0.1) | 0.500 |
| Knowledge of specific FP methods | | | | |
| Short-term | 552(78.6) | 552(78.6) | 1104(78.6) | |
| Pills | 527(75.1) | 522(74.4) | 1047(74.7) | 0.944 |
| Injectable | 507(72.2) | 514(73.2) | 1021(72.7) | 0.533 |
| Male condoms | 303(43.2) | 223(31.8) | 526(37.5) | 0.000* |
| Female condoms | 122(17.4) | 42(6.0) | 164(11.7) | 0.000* |
| Long-term | 296(42.2) | 286(40.7) | 582(41.5) | |
| IUD | 102(14.5) | 83(11.8) | 185(13.2) | 0.325 |
| Implants | 281(40.0) | 266(37.9) | 547(39.0) | 0.713 |
| Permanent | 39(5.6) | 22(3.1) | 61(4.3) | |
| Male sterilization | 16(2.3) | 11(1.6) | 27(1.9) | 0.611 |
| Female sterilization | 35(5.0) | 20(2.8) | 55(3.9) | 0.119 |
| Natural | 280(39.9) | 286(40.7) | 566 (40.3) | |
| Lactational Amenorrhea | 242(34.5) | 254(36.2) | 496(35.3) | 0.686 |
| Periodic abstinence | 93(13.2) | 102(14.5) | 195(13.9) | 0.721 |
| Withdrawal | 32(4.6) | 27(3.8) | 59(4.2) | 0.786 |

* Implies statistically significant results at 5% level of significance.

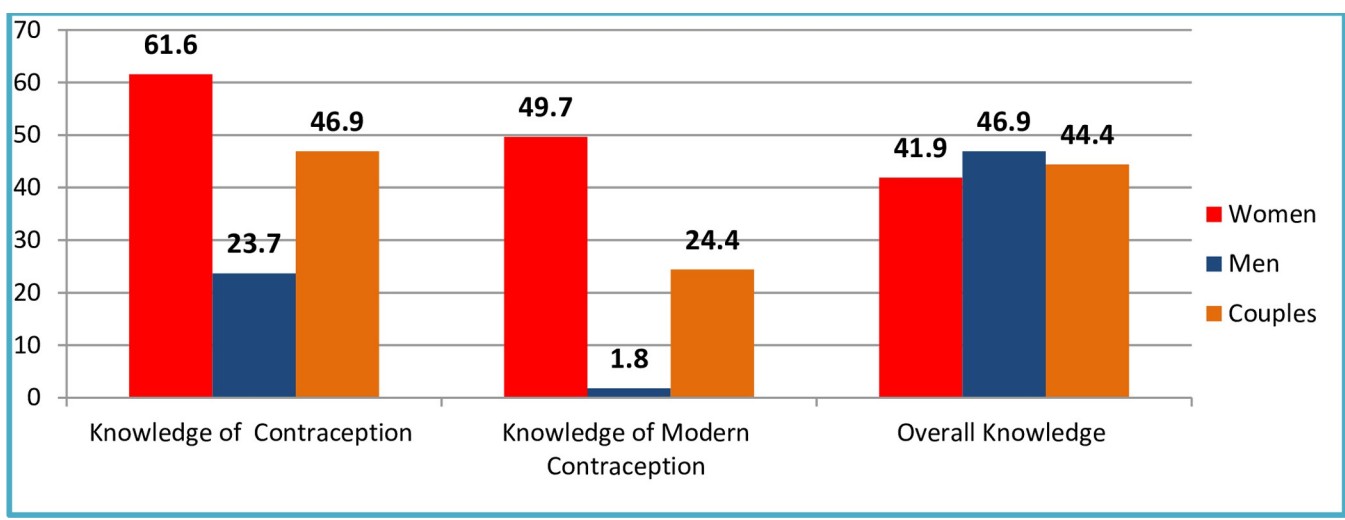

**Fig 2. Contraceptive knowledge among couples using contraception and comprehensive knowledge at study period, Fentale District, Eastern Ethiopia.**

newspaper) played a smaller role, accounting for 5.6% of the acquired information. Friends were a common source, constituting 28% for husbands and 9% for wives, as depicted in Fig 2. Other information sources included schools (15% for husbands and 13% for wives), families (15% for husbands and 1% for wives), and healthcare providers (20% for husbands and 22% for wives) (Refer to Fig 4 for details).

### 3.5. Association between contraceptive method knowledge and socio-demographic factors

Table 3 highlights a significant association between couples' education levels and contraceptive knowledge. Elevated education levels are positively correlated with increased awareness of family planning, demonstrating a consistent decrease in lower knowledge percentages—31.2% for primary education, 10.4% for secondary education, and 58.4% for those with no formal education.

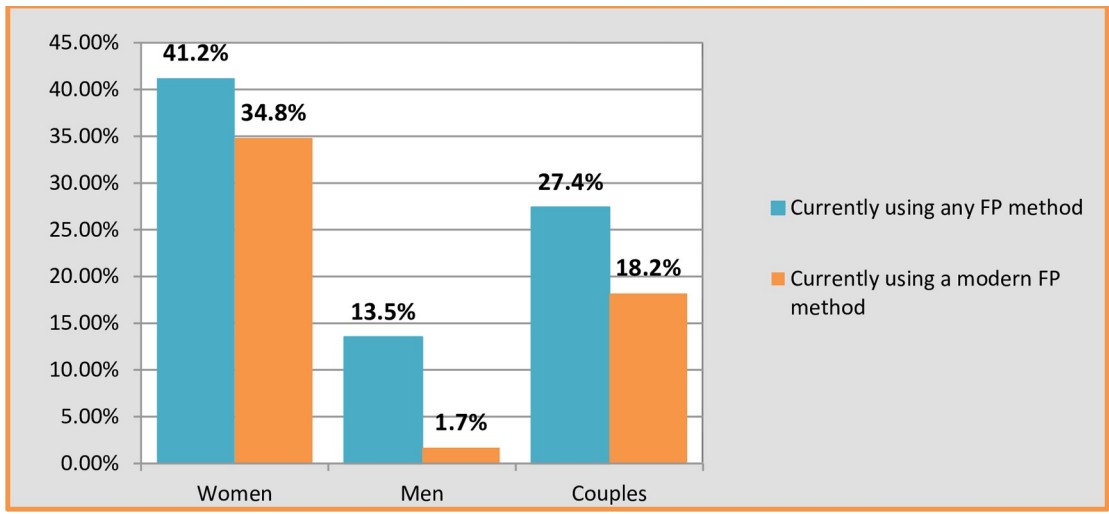

**Fig 3. Current use of modern contraceptives and any types of contraception among couples in Fentale Dictrirt, Eastern Ethiopia.**

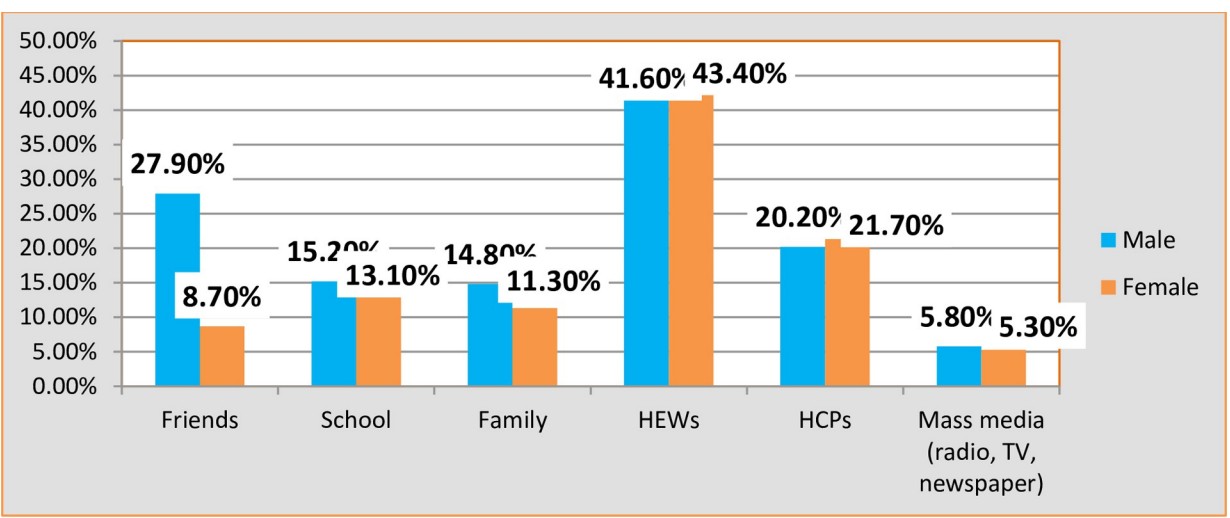

**Fig 4. Distribution of information sources among couples in Fentale District, Eastern Ethiopia, and October 1 to December 25, 2021.**

In examining media exposure, radio emerges as the predominant source, surpassing television, while minimal engagement with newspapers/magazines suggests low involvement with print media. Couples with less frequent media exposure exhibit a higher proportion of lower knowledge (89%), whereas those with more frequent exposure demonstrate a clearer understanding (11%).

Couples following a Nomadic-pastoralist lifestyle display 66.1% lower knowledge compared to Agro-pastoralists, business owners, laborers, and students. Seeking treatment from traditional healers and religious places corresponds to 2 times and 1.8 times lower knowledge than seeking treatment in the health sector, respectively. Couples without a bank account exhibit 61.7% lower knowledge compared to those with an account. Living one hour or more from a health center is associated with 60.4% lower knowledge compared to living less than an hour away (Table 3).

Binary logistic regression analysis further supports these associations, revealing that couples with primary education (AOR = 1.273) and secondary education or above (AOR = 1.642) are more likely to possess knowledge of current contraception compared to those with no formal education. Interestingly, couples engaged in business and other occupations are less likely to be knowledgeable compared to nomadic pastoralist couples. These findings underscore the importance of education and occupation in shaping contraceptive knowledge among couples (Refer to Table 3 for details).

## 3.6. Influence of Socio-demographic factors on contraceptive method awareness

Utilizing binary logistic regression, the influence of various socio-demographic factors on contemporary family planning knowledge among married couples was explored, as outlined in Table 4. Both crude and adjusted odds ratios (ORs) were analyzed to evaluate the strength of the association between predictors and knowledge of family planning as the outcome variable, after adjusting for background variables.

The results from the multivariable binary logistic regression, considering background variables, indicated that couples with primary education (AOR = 1.273; 95% CI: 1.000–1.622) and

**Table 3. Characteristics of married couples' family planning knowledge in Fentale District, Eastern Ethiopia, from October 1 to December 25, 2021.**

| Characteristics | Knowledge level(N = 1404) | |
|---|---|---|
| | **Knowledgeable**<br>**n = 623 (44.4%),** | *Less knowledgeable*<br>*n = 781(55.6%)* |
| | **Frequency (%)** | *Frequency (%)* |
| *Educational status* | | |
| *No formal education* | *300(48.2)* | *456(58.4)* |
| *Primary* | *214(34.3)* | *244(31.2)* |
| Secondary & above | *09(17.5)* | *81(10.4)* |
| *Occupational status* | | |
| Nomadic-pastoralist | *391(62.8)* | *516(66.1)* |
| Business | *44(7.1)* | *56(7.2)* |
| Others | *5(0.8)* | *20(2.6)* |
| Student | *19(3.0)* | *12(1.5)* |
| Agro-Pastoralist | *164(26.3)* | *177(22.7)* |
| *Possession of Bank account* | | |
| No | *335(53.8)* | *482(61.7)* |
| Yes | *288(46.2)* | *299(38.3)* |
| *Couple's Exposure to media* | | |
| Less Frequent | *528(84.8)* | *695(89.0)* |
| More Frequent | *95(15.2)* | *86(11.0)* |
| *Place to get Treatment or Where can sick be cured* | | |
| At Health Sectors | *174(27.9)* | *157(20.1)* |
| Traditional healers | *270(43.3)* | *339(43.4)* |
| Religious places | *170(27.3)* | *275(35.2)* |
| Others | *9(1.4)* | *10(1.3)* |
| *Distance from health center* | | |
| < 1 hour | *288(46.2)* | *309(36.6)* |
| ≥ 1 hour | *335(53.8)* | *472(60.4)* |

*Significant at α = 0.05.

secondary education or above (AOR = 1.642; 95% CI: 1.167–2.310) were more likely to have knowledge of current contraception compared to couples with no formal education.

Conversely, couples engaged in business (AOR = 0.902; 95% CI: 0.584–1.395) and other occupations, such as daily laborers and employed individuals (AOR = 0.221; 95% CI: 0.080–0.609), were less likely to be knowledgeable about current contraception compared to nomadic-pastoralist couples. However, students (AOR = 1.785; 95% CI: 0.841–3.789) and agro-pastoralists (AOR = 1.103; 95% CI: 0.853–1.427) were more likely to have knowledge of current contraception methods compared to nomadic-pastoralist couples.

Couples with bank accounts (AOR = 1.399; 95% CI: 1.108–1.766) were more likely to be knowledgeable about current contraception methods than those without bank accounts.

In terms of media exposure, radio was the most prevalent source, surpassing television, while minimal engagement with newspapers/magazines indicated low involvement with print media. Couples with more frequent media exposure (AOR = 1.333; 95% CI: 0.954–1.862) were more likely to be knowledgeable about current contraception methods than those with less frequent exposure.

Regarding treatment choices for illness, couples opting for traditional healers (AOR = 0.787; 95% CI: 0.592–1.045), religious places (AOR = 0.584; 95% CI: 0.430–0.794),

**Table 4. Multivariable logistic regression model identifying factors associated with family planning knowledge among married couples in Fentale District, Eastern Ethiopia (October 1 to December 25, 2021).**

| Characteristics | COR (95% CI | AOR (95% CI) |
|---|---|---|
| *Educational status* | | |
| *No formal education* | *1* | *1* |
| *Primary* | *0.489*(0.354, 0.675)* | *1.273*(1.000, 1.622)* |
| *Secondary & above* | *0.652*(0.463, 0.917)* | *1.642*(1.167, 2.310)* |
| *Occupational status* | | |
| *Nomadic-pastoralist* | *1* | *1* |
| *Business* | *1.037*(0.684, 1.572)* | *0.902*(0.584, 1.395)* |
| *Others* | *0.330*(0.123, 0.887)* | *0.221*(0.080, 0.609)* |
| *Student* | *2.090*(1.002, 4.356)* | *1.785*(0.841, 3.789)* |
| *Agro-Pastoralist* | *1.223*(0.952, 1.570)* | *1.103*(0.853, 1.427)* |
| *Possession of Bank account* | | |
| *No* | *1* | *1* |
| *Yes* | *1.386*(1.119, 1.716)* | *1.399*(1.108, 1.766)* |
| *Couple's Exposure to media* | | |
| *Less Frequent* | *1* | *1* |
| *More Frequent* | *1.454*(1.063,1.988)* | *1.333(0.954, 1.862)* |
| *Place to get Treatment or Where can sick be cured* | | |
| *At Health Sectors* | *1* | *1* |
| *Traditional healers* | *0.719*(0.549, 0.940)* | *0.787*(0.592, 1.045)* |
| *Religious places* | *0.558*(0.418, 0.744)* | *0.584*(0.430, 0.794)* |
| *Others* | *0.812*(0.322,2.050)* | *0.890*(0.347, 2.281)* |
| *Distance from health center* | | |
| *< 1 hour* | *1* | *1* |
| *≥ 1 hour* | *0.761*(0.615,0.942)* | *0.754*(0.601, 0.945)* |

*Significant at α = 0.05.

and other options (not seeking any treatment) (AOR = 0.890; 95% CI: 0.347–2.281) were less likely to be knowledgeable about current contraception methods compared to those seeking treatment in health sectors.

Married couples who walked one hour or more to reach the health center were 25% less likely (AOR = 0.754; 95% CI: 0.601–0.945) to be knowledgeable about current contraception methods compared to their counterparts (see Table 4).

While the adjusted odds ratio (AOR) suggests a numerical difference in contraceptive knowledge between husbands and wives (AOR = 1.016; 95% CI: 0.728–1.0420), it is important to note that the p-value associated with this comparison is 0.924. The non-significant p-value indicates that the observed difference in knowledge between men and women is not statistically significant. Therefore, we cannot conclude that there is a meaningful distinction in knowledge about current contraception methods between men and women in our study. (Refer to Table 4 for details).

## 4. Discussion

The study conducted in the Fentale District of Eastern Ethiopia provides valuable insights into family planning dynamics among nomadic pastoralist communities. Despite the challenges in

engaging this community, the study achieved an impressive 93.8% participation rate among 1496 couples [8, 17].

Distinctive age patterns at first marriage were observed, with men marrying at 18 and women at 15, reflecting cultural practices and norms within the community [29].

A significant portion of the population lacked formal education (53.8%), while a substantial number pursued nomadic-pastoral livelihoods (64.6%), illustrating the unique lifestyle and occupation of the community [30].

The community, predominantly Oromo and Muslim, emphasizes cultural foundations in family planning [14, 15]. The nomadic-pastoralist lifestyle presents challenges to information dissemination due to frequent travel [2, 31]. Media challenges affect 87.1% of couples, revealing socio-economic variations through gender-based resource disparities [32].

Health preferences vary, with 32.6% of men and 14.5% of women preferring the health sector, while 43.4% opt for traditional healers, and 31.7% choose religious leaders [33]. The study in Fentale District, Eastern Ethiopia, unveils socio-demographic insights crucial for tailoring family planning interventions to nomadic pastoralist communities, emphasizing cultural, educational, and economic factors.

The communication gap affects 93.2% of couples, and the median household size is 5, with an aspiration for 3 more children, posing challenges at the intersection of education and family size [33–35]. Recognizing these disparities is crucial for culturally sensitive family planning interventions in nomadic pastoralist communities [36].

The study highlights gender disparities in contraceptive knowledge; men know more about condoms, while women are more aware of modern methods. In Fentale District, contraceptive use among couples is 27.4%, favoring women (41.2%) over men (13.5%) [37]. Although 80.2% of couples are aware of at least one family planning method, there are gender disparities in the awareness of specific methods [8, 37–39].

The study finds a median knowledge score of 3 for both genders, with no significant statistical difference (P = 0.060) [40]. However, gender disparities emerge in awareness of specific family planning methods [39]. Men exhibit higher knowledge of male condoms (43.2%) and female condoms (17.4%) compared to women, suggesting targeted education [41], yet this did not translate into higher contraceptive use among men. This suggests that awareness alone is insufficient to drive utilization, and targeted interventions are needed to address barriers that men face in engaging with contraceptive practices [42].

Couples' awareness of male condoms is 37.5%, while female condom awareness is lower at 11.7% (p = 0.000) [41]. This underscores the need for enhanced education on a broader range of contraceptive methods, especially those designed for female use (Huber-Krum and Norris, 2020) [11, 39, 43].

While no statistically significant difference in knowledge is observed among spouses regarding short-term, long-term, and natural contraceptive methods [11, 43], there are variations in recognition levels. The most recognized methods include pills (74.7%), injectables (72.7%), and implants (39.0%) [8, 44]. However, there's a notable knowledge gap for long-term methods (41.5%), natural methods (40.3%), and permanent methods (4.3%). This suggests a preference for short-acting contraceptives, despite drawbacks like lower effectiveness and higher costs. To address this, improving the availability of diverse contraceptive methods and overcoming barriers in the Fentale pastoralist community, such as geographical distance, could promote the utilization of long-acting methods [8, 4].

These findings emphasize the need for targeted educational interventions addressing gender-specific knowledge gaps and promoting awareness of diverse contraceptive methods [38, 45]. Initiatives should strive for equitable understanding among both men and women [46,

47]. Considering the pastoralist context, interventions should be culturally sensitive and accessible, addressing the unique challenges and lifestyle of the community [8].

The study reveals a significant disparity in modern contraceptive use, with 18.2% of couples adopting it, showcasing a notable inclination of women (34.8%) compared to men (1.7%) [48]. This suggests a potential preference or accessibility gap among men for modern contraceptive options.

Among couples using modern contraception, 24.4% have awareness, with a notable difference between women (49.7%) and men (1.8%), underscoring the necessity for focused efforts to improve awareness, especially among male community members, about the benefits and choices associated with modern contraceptive methods [49, 50].

The study's quantitative findings highlight that 27.4% of married couples in the pastoralist community of Fentale District reported using any type of contraception, with notable gender disparities [37, 51]. Women exhibit a significantly higher utilization rate (41.2%) compared to men (13.5%).

The study highlights a gender gap in any type of contraceptive awareness, with 46.9% of couples using contraception showing awareness. Women exhibit significantly higher awareness (61.6%) compared to men (23.7%), emphasizing the need for targeted educational interventions to enhance contraceptive knowledge among men [42, 52].

Finally, it's important to note that Figs 2 and 3 distinctly illustrate the correlation between contraceptive knowledge and contraceptive adoption based on the findings from Fentale District. Notably, Fig 3 highlights a significant distinction: wives exhibit a higher prevalence of contraceptive usage compared to their husbands [11]. Examining Fig 2 further emphasizes that wives engaging in contraception tend to possess a superior understanding of contraceptive methods compared to their husbands. Consequently, enhancing the contraceptive uptake among couples can be achieved by augmenting their awareness of contraception [53].

The study reports an overall contraceptive knowledge of 44.4% among couples in Fentale District, Eastern Ethiopia, revealing a gender gap (41.9% for women vs. 46.9% for men). Although a marginal difference exists, it lacks statistical significance (P = 0.60), suggesting it may be due to chance. Comparisons with existing literature, including studies by [8, 50, 54], can contribute to a comprehensive understanding of gender dynamics in family planning. Exploring how other studies discuss statistical significance in gender-based differences adds value to the broader discourse on family planning knowledge [55]. The findings indicate the need for improvement in addressing gender-specific knowledge disparities and ensuring a balanced understanding among both men and women [55].

Promoting family planning in pastoralist communities like Fentale District in Eastern Ethiopia is crucial for reducing mortality, improving overall health, and fostering economic development [47]. However, challenges persist in the underutilization of modern contraceptive methods, requiring interventions to address barriers concurrently [47].

While 80.2% of couples are informed about family planning methods, the study highlights gender disparities and the need for enhanced education, especially for female-oriented methods [9, 39]. The study advocates for promoting modern contraceptive methods in pastoralist populations, aligning with global health initiatives and addressing cultural and socio-economic factors [8, 56]. The objective is to empower communities to make informed decisions for comprehensive reproductive health and align with global and national health objectives [7].

In Fentale District, Eastern Ethiopia, where contraceptive knowledge among couples, especially between men and women, has been evaluated, the practical implications of this awareness are significant [11, 53]. This understanding is crucial for tailoring interventions to meet the specific needs of the pastoralist community in Fentale District, aligning with both national

and global health initiatives [1] and contributing to the broader objectives of the national health plan [4, 8].

The study builds on existing literature [8] and emphasizes the crucial role of education in predicting contraceptive knowledge among couples. The nuanced analysis reveals a positive correlation between higher educational attainment and increased awareness of current contraception methods [11, 53]. This underscores the need for targeted educational interventions to address knowledge gaps, especially.

## 5. Limitations

Some potential limitations that warrant consideration include:

Nomadic and agro-pastoralist lifestyles in Fentale District may limit the findings' applicability to sedentary or urban populations due to unique socio-cultural dynamics.

The constant mobility of the pastoralist community poses challenges for consistent data collection, making it difficult to track individuals over time, potentially resulting in information gaps. Challenges related to language and cultural differences may impact effective communication and understanding of survey questions, affecting response accuracy.

Heavy reliance on self-reported data introduces the possibility of bias, as social desirability bias may influence participants to provide socially acceptable responses, potentially deviating from actual behaviors and knowledge. Findings suggest limited female participation and reliance on husbands for information, indicating potential constraints on female autonomy, and influencing the accuracy of reproductive health data. Cultural context challenges may hinder open discussions on certain topics, resulting in underreporting or hesitation in sharing information related to family planning practices. The nomadic nature of the community implies findings may be subject to temporal changes, capturing a specific point in time. The dynamic pastoralist lifestyle suggests evolving attitudes and practices over time.

## 6. Conclusion

In summary, the unique socio-cultural dynamics of Fentale District, characterized by its nomadic pastoralist and agro-pastoralist community, necessitate adaptive social services. Our study underscores the potential impact of couple-based health education, particularly focusing on males in predominantly male-dominated pastoralist communities like Fentale, to positively influence family planning practices. Addressing family planning disparities among married couples in pastoralist communities requires a comprehensive approach that involves both men and women, acknowledging the unique challenges they face. By prioritizing culturally informed continuous education, our study aims to contribute to improved family planning outcomes and overall reproductive health in pastoralist regions.

Promisingly, interventions or education targeted at husbands recognize the private nature of reproductive health issues and the reliance of pastoralist women on their husbands for crucial insights. The outcomes of our binary regression analysis identify specific factors requiring focused intervention, such as ownership of a bank account, media exposure, proximity to health centers, occupational status, educational background, and treatment preferences.

Engaging religious leaders and "Abbaa Gada" (Indigenous Oromo) within the predominantly Muslim population with an Oromo ethnicity emerges as a strategic initiative deeply rooted in the cultural context, with the potential to significantly enhance the effectiveness of interventions in addressing identified challenges.

The findings from Fentale District highlight the distinct socio-cultural dynamics of this nomadic pastoralist and agro-pastoralist community, where constant mobility necessitates dynamic social services. Implementing mobile clinics and educational services, facilitated by

educators well-versed in the cultural intricacies of these communities, is essential for crafting effective interventions tailored to their unique lifestyle.

In conclusion, the study offers a comprehensive understanding of family planning dynamics in nomadic pastoralist communities and identifies key areas for targeted interventions. By addressing knowledge gaps, considering socio-demographic nuances, and leveraging information sources effectively, interventions can be tailored to meet the specific needs of the pastoralist community in Fentale District, contributing to broader health objectives at national and global.

## Acknowledgments

The authors extend their appreciation to our supervisors for their valuable guidance, the diligent data collectors, and the respondents for their participation. Gratitude is also expressed to the Oromia Regional Health Bureau, Zonal authorities, and Fentale District administration for their facilitation and support throughout the study. Additionally, the authors acknowledge the 'Fantale Woreda Socio-economic Office, 2020' for providing information on the number of households in Fentale.

## Author Contributions

**Conceptualization:** Sena Adugna Beyene, Sileshi Garoma, Tefera Belachew.

**Data curation:** Sena Adugna Beyene, Tefera Belachew.

**Formal analysis:** Sena Adugna Beyene, Tefera Belachew.

**Funding acquisition:** Sena Adugna Beyene, Sileshi Garoma, Tefera Belachew.

**Investigation:** Sena Adugna Beyene, Sileshi Garoma, Tefera Belachew.

**Methodology:** Sena Adugna Beyene, Sileshi Garoma, Tefera Belachew.

**Project administration:** Sena Adugna Beyene, Sileshi Garoma, Tefera Belachew.

**Resources:** Sena Adugna Beyene, Sileshi Garoma, Tefera Belachew.

**Software:** Sena Adugna Beyene.

**Supervision:** Sena Adugna Beyene.

**Validation:** Sena Adugna Beyene, Sileshi Garoma, Tefera Belachew.

**Visualization:** Sena Adugna Beyene, Sileshi Garoma, Tefera Belachew.

**Writing – original draft:** Sena Adugna Beyene.

**Writing – review & editing:** Sena Adugna Beyene, Sileshi Garoma, Tefera Belachew.

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
