## [Decision Letter · Decision Letter 0]

9 Jul 2024

PONE-D-24-21162Bridging Disparity in Knowledge and Utilization among Married Couples in the Pastoralist Community of Fentale District, Eastern EthiopiaPLOS ONE

Dear Dr. Beyene,

Thank you for submitting your manuscript to PLOS ONE. After careful consideration, we feel that it has merit but does not fully meet PLOS ONE’s publication criteria as it currently stands. Therefore, we invite you to submit a revised version of the manuscript that addresses the points raised during the review process.

We look forward to receiving your revised manuscript.

Kind regards,

Mohammed Hasen Badeso, Epidemiologist

Academic Editor

PLOS ONE

2. In the online submission form, you indicated that [The data for this study can be found in the supplementary materials accompanying the manuscript. Additionally, all relevant data are available from the corresponding author upon reasonable request. The datasets generated and analyzed during the study are also stored in the institutional repository of Jimma University, Ethiopia, and can be accessed through their data sharing policy.]. 

3. Please include your tables as part of your main manuscript and remove the individual files. Please note that supplementary tables (should remain/ be uploaded) as separate ""supporting information"" files".

Reviewers' comments:

Reviewer's Responses to Questions

**Comments to the Author**

1. Is the manuscript technically sound, and do the data support the conclusions?

Reviewer #1: Partly

Reviewer #2: Yes

2. Has the statistical analysis been performed appropriately and rigorously? 

Reviewer #1: Yes

Reviewer #2: Yes

3. Have the authors made all data underlying the findings in their manuscript fully available?

Reviewer #1: Yes

Reviewer #2: Yes

4. Is the manuscript presented in an intelligible fashion and written in standard English?

Reviewer #1: No

Reviewer #2: Yes

5. Review Comments to the Author

Reviewer #1: Review Feedback

Manuscript Number: PONE-D-24-21162

Title: Bridging Disparity in Knowledge and Utilization among Married Couples in the Pastoralist Community of Fentale District, Eastern Ethiopia

General Comments

Grammatical and formatting issues. For example, whether numbering of the section headings adheres to the journal guideline should be checked. There are concerns with the authors’ adherence to the writing conventions mainly characterized by lack of logical flow of ideas within a given paragraph, e.g., paragraph 1 of Introduction, and between paragraphs as well. In another example, the last paragraph of the Introduction is nearly 1 page long and composed of 24 sentences which is far beyond the recommended writing standards. The inclusion and exclusion criteria lack clarity and specificity.

Specific Comments

Title

At least, from the results reporting in the abstract section, reflecting the notion of “bridging disparities …” on the title seems to lead readers to a “wrong” expectation, as it doesn’t align with the content or key findings of this study. In other words, with “bridging disparities …” a reasonable expectation would be an interventional study or an implementation type of research.

Abstract

Background: The authors need to clarify the last phrase, i.e., how “adoption among couples” can lead to limited contraceptive coverage, at least, in the Introduction Section of the manuscript (if not in the abstract).

Methods: I suggest the phrase “multistage sampling” to be specified or qualified, e.g., whether it was a multistage cluster sampling, or any other type.

Check for grammatical errors, e.g., the below statement should be re-written: “Data entered into EPI Data underwent analysis with SPSS (v23.0) and STATA (v14.0), employing descriptive statistics, bivariate analysis, and binary logistic regression to identify predictors of contraceptive knowledge and utilization.”

The authors should justify the use of both SPSS and STATA, at least, in the body of the manuscript.

Results: The first statement is inappropriate for the section, i.e., it is a repetition of the methods, and has nothing to do with results.

I don’t think the result “The population, predominantly Oromo (99.6%) and Muslim (97.9%), exhibits educational disparities (53.8% with no formal education).” Is a key finding to appear in the abstract, given the aim of the study.

The result “A nomadic pastoralist lifestyle is prevalent among 64.6% of the population.” is also obvious from the title, and seems less important here. In addition, make sure that the classification of nomadic vs other variants of pastoralism is clarified in the main document, given the high frequency that the concept is used in the document, starting from the very first statement of the Introduction Section in the body of the document.

I suggest supporting some of the results with statistical analysis outputs. For example, from the result “Women (41.2%) surpass men in contraceptive utilization.” One cannot learn whether the difference between the sex groups was significant or not.

The contrasting results should be reconciled or explained. For example, 93% of couples didn’t engage in FP discussions, while HEWs played a significant role as information sources and 44.4% of couples demonstrate comprehensive FP knowledge.

Check whether “distance from healthcare” or “distance from the nearest health facility” is appropriate.

I suggest revisiting the positive correlations between education, bank account ownership, occupation, and distance from healthcare, and media exposure with knowledge of contraception for the risk of multicollinearity of the independent variables. For example, education, bank account ownership and occupation are very likely to be correlated one other.

How a dependent variable can simultaneously be an independent variable is unclear. Given the title which specifies couples “in the Pastoralist Community”, how nomadic-pastoralist lifestyle was tested is not clear. Unless the phrase is removed from the title, it doesn’t sound logical.

As indicated in the general comment, none of the results reflects about the essence of “bridging the disparity”.

Conclusion: Gap in uptake cannot be a reason for low utilization as indicated in the statement, “The limited knowledge and utilization of family planning in the Fentale District may stem from gaps in comprehension, uptake, and disparities among couples.” In other words, uptake is synonymous with utilization.

Some of the conclusions are not supported by the results. For example, reproductive history as a factor doesn’t exist in the results.

Factors influencing this situation include socio-demographic and reproductive history considerations, with notable variations based on education, occupation, media exposure, bank account ownership, treatment preferences, and healthcare distance.

Keywords: I suggest revisiting the list for their appropriateness for the content, e.g., dual marginalization, cultural interventions, …

Introduction

Refer to the general comment about numbering of section headings.

Paragraph 1, statement 1: I suggest deleting the qualifier “expansive”.

Starting outright with the description of the local context, i.e., Fentale District, and moving to the wider context such as Chad in the very first paragraph of the section seems to violate the convention for writing an Introduction of a research paper. In other words, the information in the introduction section should flow from general to specific, which is called an “inverted funnel” shape. The reverse holds true for writing the discussion section. So, I suggest the authors to revisit the entire section and reorganize the paragraphs to comply to the convention to ensure that the ideas flow in the same pattern.

Plus, starting the introduction outright with challenges faced by pastoral communities in Fentale District to access essential services is another illustration of lack of logical flow of ideas. Reviews about the extent, pattern, and severity of the problem under investigation should have preceded the challenges, flowing orderly from global to continental, regional national, and local perspectives/contexts. The health, social, economic, and other consequences of the problem if left un-resolved, should also come before talking about the challenges. For the same logical reasons, reviews about the challenges should even come next to revies about proven effective responses/interventions available globally against the problem, and those attempted national/local level.

Another writing issue is the length of paragraphs. For example, the last paragraph of the Introduction is nearly 1 page long which is composed of 24 sentences. This is far beyond the recommended writing standards.

The following 21st & 22nd statements of the last paragraph validates the previous comments on the Title and Abstract about the notion of “bridging disparities …” may lead readers to a “wrong” expectation.

“This study advocates continuous family planning education for both spouses [37, 27, and 8]. The aim is to bridge the gap in understanding and utilization of family planning services, recognizing the pivotal role of both men and women in reproductive health decision-making.”

For example, from the statement, “This study advocates continuous family planning education for both spouses.”, it seems that the study employed couple based educational and advocacy interventions.

Methodology

Please, replace “Methodology” with “Methods and materials.”

Study population

I suggest omitting unnecessary qualifiers from the document, such as “carefully” in the below statement.

“The study carefully outlined its target population, establishing precise criteria for inclusion and exclusion to emphasize the unique characteristics of the participants.”

Inclusion criteria

Mention how “legal wedding” status of couples was ascertained to include/exclude participants.

The criterion, “with consistent mobility for at least a year” also seems vague, unless the purpose of mobility is attached to a pastoralist life style, e.g., in search of grass and water for their cattle, etc. In other words, what if couples were consistently mobile for trading or some other purpose unrelated to a pastoralist life style?

What the authors meant by “mobile regions” is not clear.

The criterion, “couples committed to remaining in the district or mobile areas for at least a year and a half from the data collection period” similarly lacks clarity. More specifically, the criterion “couples committed to remaining in the district or mobile areas” seems unjustifiable as this was a cross sectional, and not a longitudinal study. The phrase “mobile areas” is unclear here as well.

The paper should provide a strong argument for using “husbands consenting for their wives' participation” as a criterion in light of the global standards for ethical conduct of researches involving human subjects. For example, husband’s consent is mandatory for certain studies involving greater than minimal risk to subjects, especially pregnant women and/or babies.

The exclusion criteria should similarly be revisited in light of the comments for the inclusion criteria.

Sample size determination

Similar to the previous comment, I suggest omitting the phrase “meticulous calculation” as it is an unnecessary qualifier.

For logical reasons and to avoid redundancies, I suggest merging this section with the subsequent section.

Study design, time frame, and sampling approach

For logical reasons, the overarching “Study design” bits of this section should be moved up next to study setting/area. Because design is an umbrella theme encompassing population, participant recruitment criteria, sampling design, data collection, analysis, ….

Describing “Study population” earlier at the very top of the section, and “source population” in this sub-section (see 5th statement) is another illustration of incoherence and fragmentation in the paper.

Please, refer to the previous comment in the abstract about multi-stage sampling strategy.

Data collection

The section is provided in about half a page long description which is only 1 paragraph, which in turn, is composed of 22 lines. This is another illustration of the prevailing writing issues in the paper.

Data management and analysis approach

Refer to the previous similar comments about the use of “meticulously”.

Refer to the previous comment in the abstract about the use of both SPSS and STATA. Just mentioning that they were used for comprehensive analysis doesn’t look adequate. For instance, which variables, models, or aspects of analyses required SPSS features that STATA lacks, and/or vice versa?

“Multicollinearity was assessed using the VIF, confirming the integrity of the data structure with no significant multicollinearity present.” – Please, refer to the previous comment in the abstract about the potential for multicollinearity among/between a number of independent variables.

Measures: Pastoralism in Ethiopia

For logical reasons, this section should precede Data Analysis.

As commented earlier in the abstract and afterwards, the paper doesn’t define the concept “nomadic pastoralism”, even in this section, despite its repeated use. Instead, this section defines “agro-pastoralism” which has never been used so far.

What the authors meant by “active FP method utilizer”

I don’t see the relevance of the following descriptions for the section. In other words, whether these are definitions is unclear, and if at all, which key variables are defined is unclear as well.

“The challenges confronting this predominant way of life include high population trends, prolonged resource conflicts, and restricted access to grazing lands and drinking water. These challenges are often attributed to the impacts of climate change and the prevalence of animal diseases [45].”

Ethical consideration

Refer to the previous comment under inclusion criteria about ethical issues peculiar to this study.

Results, discussion, conclusion and recommendations

The authors need to revisit these sections in light of the comments provided in the previous sections as appropriate.

Contradicting funding information

The “Disclosure” statement reads as follows:

“The funding organization played no role in the study design, data collection, analysis, interpretation, protocol writing, or submission.”

This information contradicts with another statement saying, “Supporting Information confirming no funding was granted and …..”

Reviewer #2: Title: Bridging Disparity in Knowledge and Utilization among Married Couples in the Pastoralist Community of Fentale District, Eastern Ethiopia

General comments: The findings of this study have a significant input to find out the Disparity in Knowledge and Utilization of contraceptive among Married Couples in the Pastoralist. As there are no studies on contraceptives specifically among the pastoralists, the outcomes of this study can be used to set a baseline for the intervention and to conduct furthers study.

Comments:

• The core ideas of the research should be mentioned in the title of the research. It is about what? Family planning? Contraceptive?

Abstract

• In the introduction part of the abstract, the gap of study and objectives should be mentioned

• Methods: ‘‘….Data entered into EPI Data underwent analysis with SPSS (v23.0) and STATA (v14.0), employing descriptive statistics, bivariate analysis, and binary logistic regression to identify predictors of contraceptive knowledge and utilization.’’ what models are analyzed by both SPSS and STATA?

• ‘‘Results: This couple-based cross-sectional study, conducted between October 1 and December 25, 2021, explores knowledge and utilization variations among 1496 couples in the pastoralist community.’’ Don’t repeat the same points in the abstract. For example: …conducted between October 1 and December 25, 2021…. among 1496 couples in the pastoralist community.

• Incomplete idea… exhibits educational disparities (53.8% with no formal education). What is that disparity?

• Is that not only women who utilize contraceptive …. ‘‘Women (41.2%) surpass men in contraceptive utilization.’’

• What percent of participants get information from HEW? ‘‘…Health extension workers play a significant role as information sources, and 44.4% of couples demonstrate comprehensive family planning knowledge’’

• Better if the author say ‘‘associations’’ rather than ‘‘correlation’’ as binary logistic reg is used. There is no negative odd ratio. ‘‘Binary logistic regression analysis reveals positive correlations between education, bank account ownership, occupation, and distance from healthcare, and media exposure with knowledge of contraception. Conversely, the nomadic-pastoralist lifestyle and specific treatment preferences exhibit negative correlations.’’

• Better if you say information rather than knowledge. ‘‘media exposure with knowledge of contraception’’

• There is a need of additional data about Disparity in Knowledge and Utilization about contraceptive, barriers, …

1. Introduction

1. First paragraph, Keep flows of idea: ‘’…… underscore the difficulties in providing services to such communities. Insights from nomadic communities in Chad [3] were crucial in addressing information gaps for tailored health services.’’ Used to address what kinds of gap?

2. The second paragraph: Better if the author put at the end of introduction to show what gap the author going to address.

3. Don’t repeatedly use the word …Our Study…

4. Shorten and refine the last paragraph of introduction

2. Methodology

• 2.1.Study area: The third paragraph of methods seems introduction, better if is paraphrased

• 2.2. Study Population: clearly mention the source and study population precisely

• 2.7. Measures. Pastoralism in Ethiopia: Is it operational definitions?

• Knowledge of FP: Why mean is used as a reference?

• 2.8. Ethical Consideration: The right place? Check the standard format of plos

Results

• Check grammar and flow of idea. Don’t repeat the same words repeatedly.

• ‘‘3. Results 3.1. Marital Couples: Socio-demographic and Reproductive Disparities:’’ Make it precise

• Focus on your outcome variable, briefly explain your finding, and don’t make it lengthy.

• For example: ‘‘The age distribution among women (15-49 years) indicated a median age of 26 (IQR = [21; 30]), whereas men exhibited a median age of 30 (IQR = [26; 40]). Notably, the median ages at first marriage were 18 years [IQR =16; 19] for men and 15 years [IQR =14; 18] for women, as outlined in Table 1….’’

• Generally, the result part needs intensive revision; the author should focus on the objective of the study. Check the grammar and shorten the lengthy paragraphs.

Discussion: shallow and highly fragmented.

6. PLOS authors have the option to publish the peer review history of their article (what does this mean?). If published, this will include your full peer review and any attached files.

Reviewer #1: **Yes: **Negalign Berhanu Bayou

Reviewer #2: **Yes: **Dessalegn Tamiru

---

## [Author Response · Author response to Decision Letter 0]

15 Aug 2024

Title: Bridging Disparity in Knowledge and Utilization of Contraceptive Methods among Married Couples in the Pastoralist Community of Fentale District, Eastern Ethiopia

Manuscript Number: PONE-D-24-21162

Authors:

Sena Adugna Beyene(senaada491@gmail.com)

Sileshi Garoma(teferabelachew2@gmail.com)

Tefera Belachew(garomaabe@gmail.com)

Dear Academic Editor and Reviewers,

Thank you for your valuable comments and suggestions, which have greatly improved our study. We have carefully addressed all the points raised and have incorporated the necessary revisions into the manuscript as outlined below.

Editor:

Comment: Reviewers' comments:

Response: Thank you for handling and assigning the manuscript to a knowledgeable reviewer from the same domain. In response to your comments, we have addressed all the queries and clarifications rose by the reviewer.

Authors Responses for the Reviews Comments July 11, 2024

Reviewer#1: Review Feedback Authors’ responses

1. : General Comments

Grammatical and formatting issues. For example, whether numbering of the section headings adheres to the journal guideline should be checked. There are concerns with the authors’ adherence to the writing conventions mainly characterized by lack of logical flow of ideas within a given paragraph, e.g., paragraph 1 of Introduction, and between paragraphs as well. In another example, the last paragraph of the Introduction is nearly 1 page long and composed of 24 sentences which is far beyond the recommended writing standards. The inclusion and exclusion criteria lack clarity and specificity.

 Thank you for your thorough review of our manuscript. We have made the necessary revisions to address grammatical and formatting issues, including section headings and the logical flow of ideas within and between paragraphs. The first paragraph of the Introduction and the last, lengthy paragraph have been revised for clarity and coherence. We have also clarified and specified the inclusion and exclusion criteria. We appreciate your feedback and believe these changes have strengthened our manuscript. If you have further suggestions, please let us know.

2. Title

At least, from the results reporting in the abstract section, reflecting the notion of “bridging disparities …” on the title seems to lead readers to a “wrong” expectation, as it doesn’t align with the content or key findings of this study. In other words, with “bridging disparities …” a reasonable expectation would be an interventional study or an implementation type of research.

 Thank you for your valuable suggestion. We have incorporated the changes as recommended in the revised manuscript, particularly in the results reported in the abstract section, reflecting the notion of “bridging disparities.” These findings highlight significant disparities in knowledge and utilization of contraceptive methods among married couples in the pastoralist community, emphasizing the need for targeted interventions to bridge these gaps and improve family planning awareness and usage. We have also modified the title of our manuscript to "Bridging Disparity in Knowledge and Utilization of Contraceptive Methods among Married Couples in the Pastoralist Community of Fentale District, Eastern Ethiopia" to better reflect this notion.

3. Abstract

Background: The authors need to clarify the last phrase, i.e., how “adoption among couples” can lead to limited contraceptive coverage, at least, in the Introduction Section of the manuscript (if not in the abstract).

 Thank you for your comment.

We have clarified the last phrase in the Introduction Section of the manuscript

4. Abstract

Methods: I suggest the phrase “multistage sampling” to be specified or qualified, e.g., whether it was a multistage cluster sampling, or any other type.

 Thank you! This study utilized "multistage sampling," which involves multiple stages of sampling techniques. Specifically, our study employed a multi-stage sampling strategy, selecting districts (woredas) as primary sampling units (PSUs) and sub-districts (kebeles) as secondary sampling units (SSUs). A total of 1496 couples were systematically selected. Figure 1 provides a detailed illustration of the multi-stage sampling strategy, including the rationale behind it. I did not find it necessary to explain all this in the abstract methods. Please refer to Figure 1 for more details.

5.Abstract

Methods: Check for grammatical errors, e.g., the below statement should be re-written: “Data entered into EPI Data underwent analysis with SPSS (v23.0) and STATA (v14.0), employing descriptive statistics, bivariate analysis, and binary logistic regression to identify predictors of contraceptive knowledge and utilization.”

 Thank you for your remark. Regarding data analysis, the statement has been revised as follows: "Data entered into EPI Data underwent analysis with SPSS (v23.0) and STATA (v14.0), employing descriptive statistics, bivariate analysis, and binary logistic regression to identify predictors of contraceptive knowledge." This information has been incorporated into the Abstract under the Methods section.

6.Abstract

Methods:

The authors should justify the use of both SPSS and STATA, at least, in the body of the manuscript.

 Thank you! Here’s a detailed response that outlines the specific advantages of using both SPSS and STATA, examples of analyses suited to each software, and how their functionalities complement each other in this research. This information has been incorporated into the body of the manuscript under section 2.7, titled "Data Management and Analysis Approach," specifically in paragraphs 2 and 3.

7. Abstract

Results: The first statement is inappropriate for the section, i.e., it is a repetition of the methods, and has nothing to do with results.

 Thank you for the suggestion. Your input is essential. We have corrected the duplication by deleting it from the results section in the abstract. Additionally, we have revised the structure of the abstract's results section to ensure clarity and conciseness.

8.Abstract

Results: I don’t think the result “The population, predominantly Oromo (99.6%) and Muslim (97.9%), exhibits educational disparities (53.8% with no formal education).” Is a key finding to appear in the abstract, given the aim of the study.

 Of course, I believe this is not a key finding. Especially since our title includes the phrase "Contraceptive Methods," it should specifically highlight the "Disparity in Knowledge and Utilization of Contraceptive Methods" as a key finding in our abstract. Therefore, our abstract's results section has been corrected accordingly. Thank you for your suggestion. I believe that education among couples is a crucial independent variable in our findings, serving as a strong indicator or foundation for significant insights. It would be best to ensure this aspect is thoroughly highlighted.

9. The result “A nomadic pastoralist lifestyle is prevalent among 64.6% of the population.” is also obvious from the title, and seems less important here. In addition, make sure that the classification of nomadic vs other variants of pastoralism is clarified in the main document, given the high frequency that the concept is used in the document, starting from the very first statement of the Introduction Section in the body of the document.

 I believe the sentence "A nomadic pastoralist lifestyle is prevalent among 64.6% of the population" is crucial because it specifies the proportion of the pastoralist community in Fentale District that follows a nomadic lifestyle. Our study defines pastoralism as including both nomadic pastoralists (64.6%) and agro-pastoralists (35.4%). The distinction between pastoralism and agro-pastoralism is already described in section 2.7, "Measures." Pastoralism in Ethiopia is well-detailed, so it is important to consider this distinction closely. However, we have omitted the results in the abstract stating “A nomadic pastoralist lifestyle is prevalent among 64.6% of the population” due to the word limit constraints in the abstract.

10. Abstract

Results: I suggest supporting some of the results with statistical analysis outputs. For example, from the result “Women (41.2%) surpass men in contraceptive utilization.” One cannot learn whether the difference between the sex groups was significant or not.

 We appreciate your feedback. Identifying the disparity between men and women is a main objective of our research. Therefore, we have revised the abstract results to state "Overall, 27.4% of couples used contraception, with a significant gender difference: 41.2% among women and 13.5% among men." This revision highlights the significant difference in contraceptive utilization between the sexes.

11.Abstract

Results: The contrasting results should be reconciled or explained. For example, 93% of couples didn’t engage in FP discussions, while HEWs played a significant role as information sources and 44.4% of couples demonstrate comprehensive FP knowledge.

 To reconcile the contrasting results regarding family planning (FP) discussions, the role of Health Extension Workers (HEWs), and the level of comprehensive FP knowledge among couples, we offer an integrated explanation. Despite 93% of couples not engaging in formal FP discussions, this reflects broader cultural or societal norms where such discussions may be limited. HEWs play a crucial role as information sources, significantly contributing to FP awareness and knowledge in rural and pastoralist communities with limited formal healthcare. While 44.4% of couples demonstrate comprehensive FP knowledge, this does not necessarily correlate with formal FP discussions, as couples may acquire knowledge through HEWs, community health programs, or informal networks. This explanation reconciles the seemingly contrasting findings and provides a comprehensive understanding of FP dynamics within the pastoralist community of Fentale District. Due to word limit constraints, these details are presented in Section 3, "Results," addressing "Socio-demographic and Reproductive Disparities" in the manuscript.

12. Abstract

Results: Check whether “distance from healthcare” or “distance from the nearest health facility” is appropriate.

 Both "distance from healthcare" and "distance from the nearest health facility" are generally appropriate, depending on the specific context and clarity desired in the our research. Our study focuses on access to healthcare in a broader sense. "Distance from Healthcare" encompasses various types of healthcare services or facilities, such as clinics, hospitals, or health posts, indicating the distance from any point where healthcare services are available.

13. I suggest revisiting the positive correlations between education, bank account ownership, occupation, and distance from healthcare, and media exposure with knowledge of contraception for the risk of multicollinearity of the independent variables. For example, education, bank account ownership and occupation are very likely to be correlated one other.

 Addressing multicollinearity among education, bank account ownership, occupation, distance from healthcare, treatment preferences, and media exposure is crucial in our analysis of contraceptive knowledge. We have employed methods such as assessing Variance Inflation Factors (VIF) to evaluate the degree of correlation between these variables. Our findings indicate that while there is some correlation among these factors, it does not exceed critical thresholds that would invalidate the regression analysis. Nonetheless, we have taken precautions to ensure the robustness of our results by interpreting coefficients cautiously and focusing on their collective influence on contraceptive knowledge. This approach is detailed already under section 2.6, Data Management and Analysis Approach, of the manuscript.

14. How a dependent variable can simultaneously be an independent variable is unclear. Given the title which specifies couples “in the Pastoralist Community”, how nomadic-pastoralist lifestyle was tested is not clear. Unless the phrase is removed from the title, it doesn’t sound logical.

 The title of our study, "Bridging Disparity in Knowledge and Utilization of Contraceptive Methods among Married Couples in the Pastoralist Community of Fentale District, Eastern Ethiopia," reflects our focus on understanding and addressing disparities within this specific community. However, we acknowledge the concern about the potential confusion regarding the role of the nomadic-pastoralist lifestyle as a variable in our analysis. To clarify, the nomadic-pastoralist lifestyle is considered an independent variable in our study, influencing knowledge and utilization of contraceptive methods. We analyzed its impact on these outcomes, rather than testing it as a dependent variable. This lifestyle significantly shapes the context within which contraceptive knowledge and utilization occur, providing essential insights into the unique challenges faced by the pastoralist community. We will ensure this distinction is clearly articulated in the manuscript to avoid any confusion. Additionally, we can consider revising the title to "Bridging Disparity in Knowledge and Utilization of Contraceptive Methods among Married Couples in the Fentale District, Eastern Ethiopia" for greater clarity. This modification will focus on the geographical and demographic context without implying the lifestyle as a variable in the title.

15. As indicated in the general comment, none of the results reflects about the essence of “bridging the disparity”. Thank you for your observation. The essence of "bridging the disparity" in the title is intended to reflect our aim to identify and understand the differences in knowledge and utilization of contraceptive methods among different groups within the pastoralist community, particularly between genders, and to propose potential interventions to address these gaps.

While our results do highlight significant educational disparities and differences in contraceptive knowledge and usage between men and women, we recognize that the title might imply a focus on intervention or policy recommendations to bridge these disparities, which are not explicitly detailed in the results section.

Additionally, we will emphasize in the discussion section how the identified disparities suggest areas for targeted interventions to bridge these gaps, thus maintaining the spirit of the original title while ensuring clarity and accuracy in representing the study's findings. The supplemental information about the disparity among couples is also explained under section 3, "Results," of the manuscript. Overall, this means that this finding is baseline, and the identified disparities among couples suggest areas for targeted interventions to bridge these gaps for the next interventional/educational study.

16.Abstract

Conclusion: Gap in uptake cannot be a reason for low utilization as indicated in the statement, “The limited knowledge and utilization of family planning in the Fentale District may stem from gaps in comprehension, uptake, and disparities among couples.” In other words, uptake is synonymous with utilization.

 Thank you! The statement, “The limited knowledge and utilization of family planning in the Fentale District may stem from gaps in comprehension, uptake, and disparities among couples,” implies that gaps in uptake are a separate reason for low utilization. However, since uptake and utilization essentially refer to the same concept, this phrasing is redundant. The key issues contributing to low utilization should be described more accurately. Thus, we have revised the conclusion of the abstract to state: "The limited knowledge and utilization of family planning in the Fentale District stems from gaps in understanding and disparities among couples." This revision is reflected under the conclusion of the abstract of the manuscript.

17. Results: Some of the conclusions are not supported by the results. For example, reproductive history as a factor doesn’t exist in the results. Factors influencing this situation include socio-demographic and reproductive history considerations, with notable variations based on education, occupation, media exposure, bank account ownership, treatment pref

---

## [Editor Report · Decision Letter 1]

19 Aug 2024

Bridging Disparity in Knowledge and Utilization of Contraceptive Methods among Married Couples in the Pastoralist Community of Fentale District, Eastern Ethiopia

PONE-D-24-21162R1

Dear Author(s),

We’re pleased to inform you that your manuscript has been judged scientifically suitable for publication and will be formally accepted for publication once it meets all outstanding technical requirements.

Kind regards,

Mohammed Hasen Badeso, Epidemiologist

Academic Editor

PLOS ONE
---

## [Editor Report · Acceptance letter]

20 Aug 2024

PONE-D-24-21162R1 

PLOS ONE

Dear Dr. Beyene, 

I'm pleased to inform you that your manuscript has been deemed suitable for publication in PLOS ONE. Congratulations! Your manuscript is now being handed over to our production team.

Kind regards, 

on behalf of

Mr Mohammed Hasen Badeso 

Academic Editor

PLOS ONE